# Top-Leader Growth in Nordmann Fir (*Abies nordmanniana*)

**Bjarke Veierskov**

Department of Plant and Environmental Sciences, Transport Biology, University of Copenhagen, Thorvaldsensvej 40, 1871 Frederiksberg C, Denmark; bv@plen.ku.dk

**Abstract:** The growth of the top-leader in *Abies nordmanniana* was measured over a 6-year period, and showed a consistent growth pattern, with an initial growth phase reaching a maximum growth rate that varied between 12.0 and 22.5 mm per day, and which could be correlated with the temperature in July of the previous year. The daily growth rate always peaked in the last week of June. In late-bud-breaking plants, the top-leader became short because of a low growth rate in this first phase of growth. In the second growth phase, the daily growth rate slowly declined, but was similar in all plants, regardless of the size of the top-leader when this phase began, and the timing of the bud break. The growth in the second growth phase was correlated with the precipitation in the period. Growth cessation occurred in the basal part of the top-leader soon after bud break, and progressed slowly apically, whereby the upper 25% of a young top-leader contributed to 50% of the final length, as growth in this section continued throughout the major part of the growth season.

**Keywords:** *Abies nordmanniana*; growth rate; shoot development; stem growth

## 1. Introduction

The Nordmann fir is the main Christmas tree grown in Europe, and until the beginning of the twentieth century production was based primarily on natural trees [1]. Since then, consumer quality requirements for a Christmas tree have increased, and today production is intensively administered by weed control, pest management, and fertilisation, as well as top-leader control and trimming of sidebranches [2]. The economic return of this high-value crop therefore depends on the grower's ability to shape the trees to demand of consumers. The length of the top-leader is one of the important quality factors. If not treated, the top-leaders can easily grow to more than 55 cm tall, well above the preferred 30–35 cm [2]. Obtaining a predetermined length for the top-leader by chemical or mechanical treatments requires a detailed knowledge of the growth pattern. In *Abies nordmanniana*, buds that contain the entire number of stem units are formed from July to November, and elongation occurs in spring of the following year [3,4]. Annual growth begins when all stem units in the primordial shoot begin to elongate simultaneously [5], and in conifers elongation is caused by reactivation of cell division in the rib zone [6]. In an elongated stem, the driving force for cell elongation originates from the internal turgor pressure, primarily in the pith cells, while the epidermal cells restrict growth [7]. To reach the end-of-season length of the top-leader, additional cell divisions must occur within the stem units [4]. In 1845, Harting observed that differences in internode length in Tillia parvi-flora L. were mainly due to differences in cell number (cited in [8]), which was later confirmed in several tree species by Brown and Somme [8], who further showed that, although genetics can regulate cell length in different apple cultivars, cell number was still the most important factor in determining the length of internodes [9].

The increasing impact of climate change on forest productivity has been evident in the last 50 years [10]. Climate–growth relationships such as temperature and precipitation/drought are climatic factors with a great impact on tree growth and development such as shoot growth [11–16], but also on stem growth [17,18], bud break, and increased leaf production [10,15,19].



In the 1940s and 1950s, several papers described the interaction between climate and elongation of the top-leaders of trees, reviewed by Kozlowski in 1964 [20], an area that has regained attention in relation to the global climate changes [15,18,19,21]. As the growth cycle in *Abies nordmanniana* is divided into two growing seasons [3,4], it is important to evaluate the interaction of growth with climate conditions not only in the current year of elongation of the top-leader, but also in the previous season, when the primordial shoot is formed.

As Christmas trees must be shaped according to consumers demand, it has become an increasing resource requirement for the growers to shape the trees accordingly. To be able to imply optimal treatment at the most efficient time of the tree's development, an improved understanding of the the physiology of top-leader elongation is a prerequisite for controlling its elongation in the Nordmann fir. To obtain a better understanding of top-leader elongation, top-leader elongation was monitored in uniform 8-year-old trees over a 6-year period, and possible correlations with climatic conditions were determined.

## 2. Materials and Methods

### 2.1. Site Description

Measurements were made in the years 2016 to 2022. To obtain plants of equal developmental size each year, measurements were made on trees grown on two different commercial Christmas tree farms where similar trees were grown. In 2016–2019, the experiments were carried out on a 5 ha farm located in Jutland at Starupvej, 8340 Malling (N56.0276, E10.20587), while the experiments in 2020–2022 were carried out on an 8 ha farm located in Zealand at Røglevej, 3540 Lynge (N55.8558, E12.2948). Both farms have similar soil types (moraine loam) and a plant density of 6000 trees per ha. The trees were fertilized with a commercial fertilizer, with 50 kg N ha (NPK 23-3-7 + Mg), split in two, two thirds in April and one third in July [22].

### 2.2. Climate Data

The plants were grown under ambient climatic conditions. The monthly average of weather data including the min, max and average temperature, precipitation, sunshine hours, and drought index was obtained from the Danish Meteorological Institute weather stations closest to the Christmas tree farms (https://www.dmi.dk/vejrarkiv/, accessed on 10 November 2022). The data are presented in Supplementary Data Table S1. In 2022, the temperature, precipitation, and irradiance was monitored every 30 min by a weather station located on the Christmas tree farm (HoBo X3000 Remote Monitoring Station).

### 2.3. Plant Material

The plant material was *Abies nordmanniana* Spach prov. Ambrolauri. Each year, in April, from 2016 to 2021, uniform 8-year-old trees between 1.2 and 1.5 m tall with top-leaders from the previous year of 30 to 40 cm were selected. The plants were selected from among 10,000 trees, and thus constituted less than 1% of the population. Measurements were made in the years 2016 to 2022. In 2016 to 2021, the selected trees were randomly divided into 3–5 lots. In 2016, 40 trees were divided into 4 lots of 10 trees; in 2017, 60 trees were divided into 4 lots of 15 trees; in 2018, 30 trees were divided into 3 lots of 10 trees; in 2019, 21 trees were divided into 3 lots of 7 trees; in 2020, 75 trees were divided into 5 lots of 15 trees; and in 2021, 15 trees were divided into 3 lots of 5 trees. In 2022, tree selection occurred on 22 June, when trees having top-leaders of 5 cm (2 lots of 5 trees), 10 cm (4 lots of 5 trees), 15 cm (5 lots of 5 trees) or 20 cm (3 lots of 5 trees) were selected.

### 2.4. Measurements

Each year, the length of the top-leaders of the individual trees was determined three times weekly (with a 2- or 3-day interval), from bud-break until growth ceased in August. However, in 2020, measurements were made daily. For additional growth analysis among the entire group of 75 trees in 2020, a special tag was put on the fastest 6 bud-bursting

plants (selected for having top-leaders of 50 mm on 2 June, and with the slowest 6 plants having 50 mm long top-leaders on 15 June).

The top length was measured from the top of the northernmost branch of the first whorl to the tip of the top-leader. Section measurements were made in 2017, when 15 top-leaders of approximately 200 mm were divided into 3 groups of 5, and the top-leaders were divided into 4 sections on 15 June (1 to 4, numbered from the base of the top-leader) using a marker pen. The three upper sections were all 50 mm long, while the length of the lower section varied between 40 and 50 mm. At the time the maximal growth rate was reached, in 2022 (22 June), top-leaders of either 5, 10, 15 or 20 cm tall were selected.

### 2.5. Statistical Data Analysis

At each measurement date for each year, the mean values of individual lots were determined and used for statistical analysis, and Student's *t* test was performed to identify significant differences. The average of each group was calculated $\pm$ SD. The significance of the growth curves was determined by an ANOVA test. The correlations were established according to the significance of the Pearson correlation coefficient.

## 3. Results

### 3.1. Top-Leader Elongation

During all the years from 2016 to 2022, bud burst occurred over a 2-week period in late May/early June, and growth continued until 15 July, when 95% of the final length was obtained (Figure 1A). The exception was 2018, when growth began about 2 weeks earlier, but growth progressed similarly to the average of the period 2016–2022, excluding 2018, and 95% of the final length was reached on 4 July, 11 days earlier than the average of the other years. However, in 2018, the temperature in April and May in Denmark was 1.7 and 3.8 °C warmer, respectively, than the average during the rest of the years 2016–2022 (Table 1).

**Table 1.** Average monthly temperature in the Christmas farms at Malling (2016 to 2019) and Lynge (2020–2022). Data were obtained from the Danish Meteorological Institute (https://www.dmi.dk/vejrarkiv/).

| | | | | | | | | | | | | °C |
|---|---|---|---|---|---|---|---|---|---|---|---|---|
| **Year** | **April** | **May** | **June** | **July** | **August** | **September** | **October** | **November** | **December** | **January** | **February** | **March** |
| 2016 | 6.0 | 12.7 | 15.8 | 16.1 | 15.8 | 15.8 | 8.9 | 3.9 | 4.8 | 0.1 | 2.1 | 3.7 |
| 2017 | 6.2 | 11.9 | 14.5 | 15.3 | 15.7 | 12.8 | 10.7 | 5.0 | 3.3 | 1.2 | 1.9 | 4.7 |
| 2018 | 8.3 | 14.9 | 16.4 | 19.4 | 17.2 | 13.6 | 9.8 | 5.8 | 4.1 | 2.1 | −0.9 | 0.1 |
| 2019 | 7.5 | 9.4 | 15.8 | 16.6 | 16.9 | 13.0 | 9.0 | 5.8 | 4.6 | 1.4 | 4.3 | 5.1 |
| 2020 | 7.8 | 10.3 | 16.9 | 15.4 | 18.5 | 14.1 | 10.2 | 7.5 | 4.2 | 5.0 | 4.5 | 4.4 |
| 2021 | 5.7 | 10.5 | 17.1 | 19.1 | 15.7 | 14.4 | 10.4 | 6.7 | 1.6 | 0.5 | −0.1 | 3.9 |
| 2022 | 6.3 | 11.8 | 15.6 | 17.1 | 18.7 | 13.3 | 11.7 | 7.4 | 1.1 | 3.5 | 3.7 | 3.6 |

Dividing 20 cm tall top-leaders into four equal sections revealed that in section 1 elongation had ceased by 20 June, and in section 2 growth ceased 4 days later, while growth continued for an additional 2 weeks until 7 July in section 3. However, in the upper section 4, the elongation continued until 6 August (Figure 2A). Sections 1, 2, 3 and 4 contribute to the final length, with 66 mm (12%), 88 mm (16%), 133 mm (24%), and 270 mm (48%), respectively. Thus, the upper 25% of the top-leader contributes half of the final length of the top-leader.

Monitoring the top-leaders who had an early (24 May) or late (7 June) bud break revealed that a late bud break reduced the final length of the top-leader by 47%, compared to early breaking buds (299 vs. 562 mm) (Figure 3A), but both had reached 95% of the final length on 20 June (Figure 4). This growth pattern was confirmed by following the growth of the top-leaders, which were 5, 10, 15 or 20 cm long on 22 June (Figure 4A).

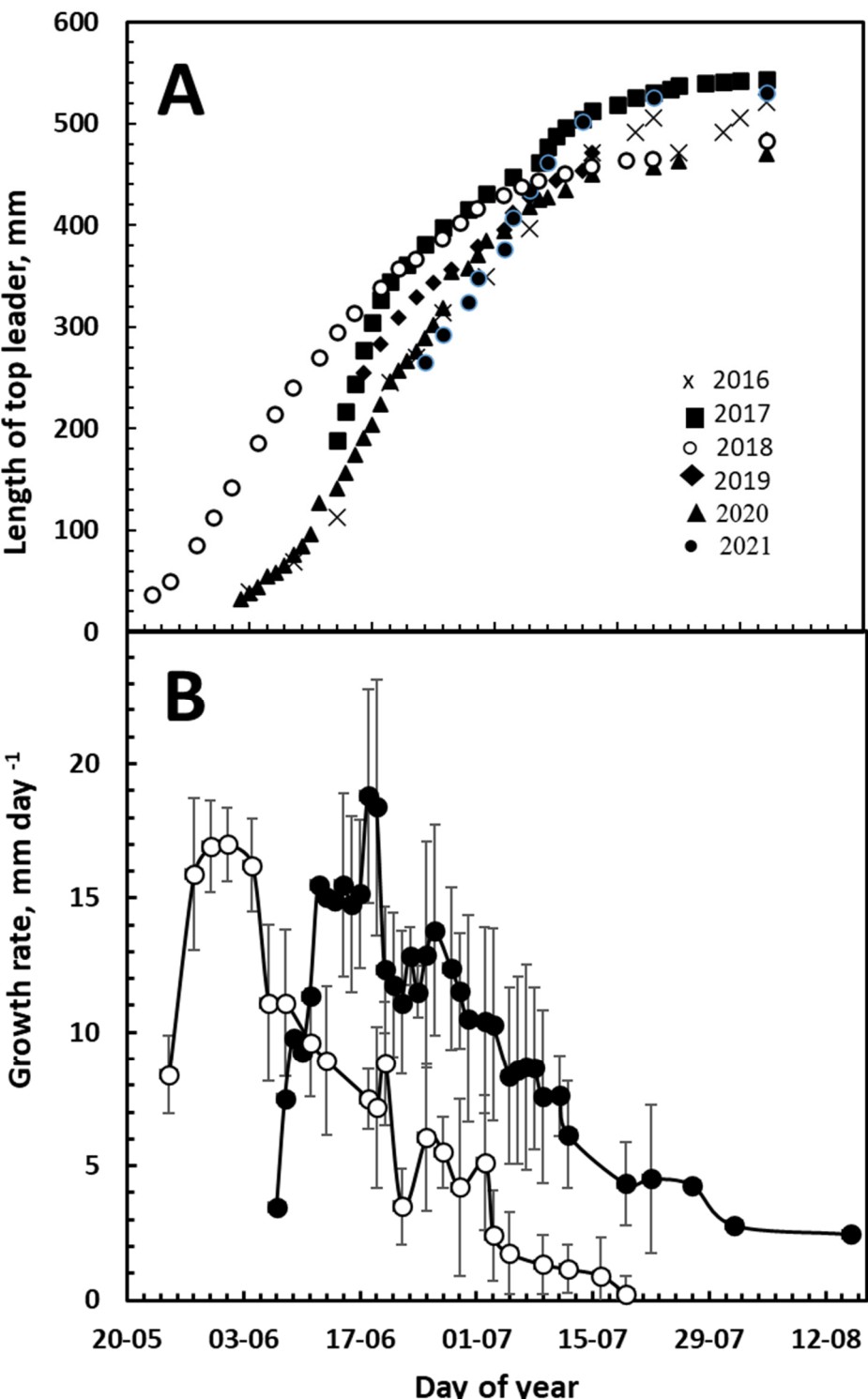

**Figure 1.** Top-leader growth in Abies nordmanniana. Length of top-leader (**A**) and daily growth rate 2016–2021. (**B**) Daily growth rate given as means of averages for the years 2016, 2017, 2019 and 2020 (●) or in the 2018 growing season (○). Data in (**B**) are presented as means ± SD.

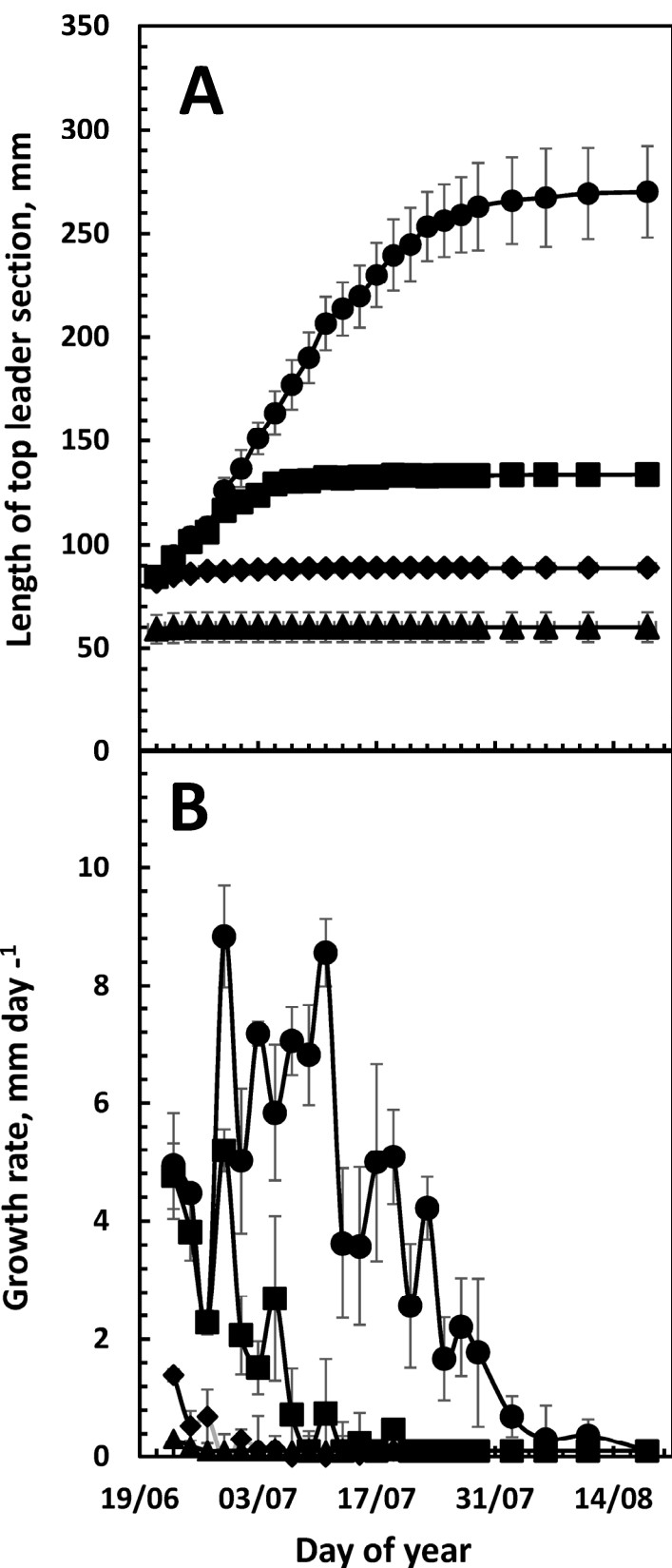

**Figure 2.** Growth of the individual sections of the top leader: Section 1 (▲), Section 2 (♦), Section 3 (■) or Section 4 (●) numbered from the base of the top-leader. (**A**) Length of the individual sections. (**B**) Daily growth rate of the individual sections. Each section constituted 25% of the top length on 15 June. The growth of each section was followed until 19 August. Data are from the 2020 season and given as means of four replications of ten plants each. Data given as mean ± SD.

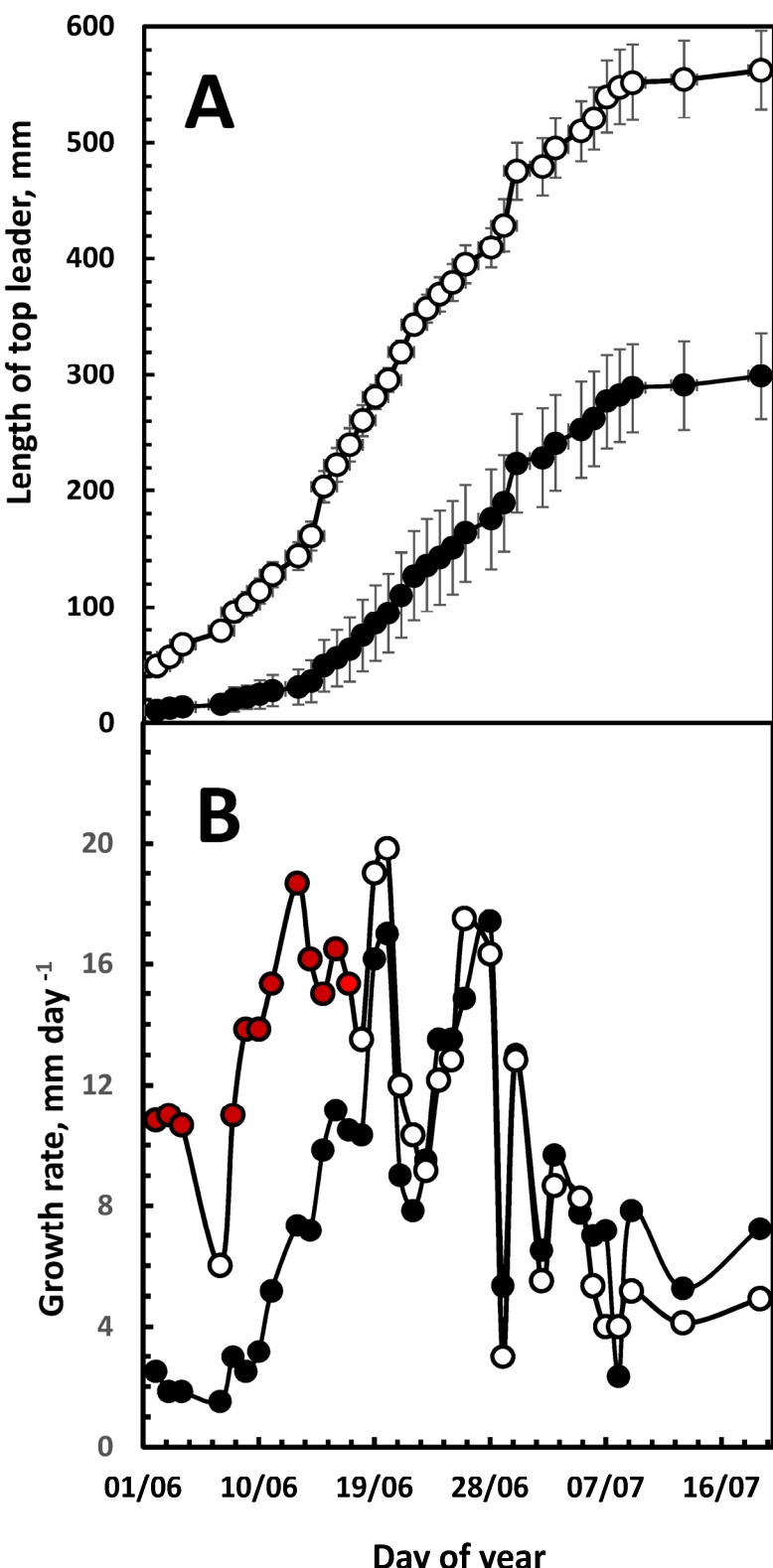

**Figure 3.** Growth of the top-leaders of plants having an early (○) or late (●) bud burst. (**A**) Length of the top leader. (**B**) Daily growth rate of the top leader. The plants were selected from the 2020 growth season. Early-bud-burst plants had top-leaders of 50 mm on 2 June, whereas the length was reached on 15 June in the late-bud-bursting plants. Each group consisted of 6 plants, and growth was determined daily. The points of significant differences are marked by red filling-in in the early bud-bursting data points. The length of the leaders is given as means ± SD.

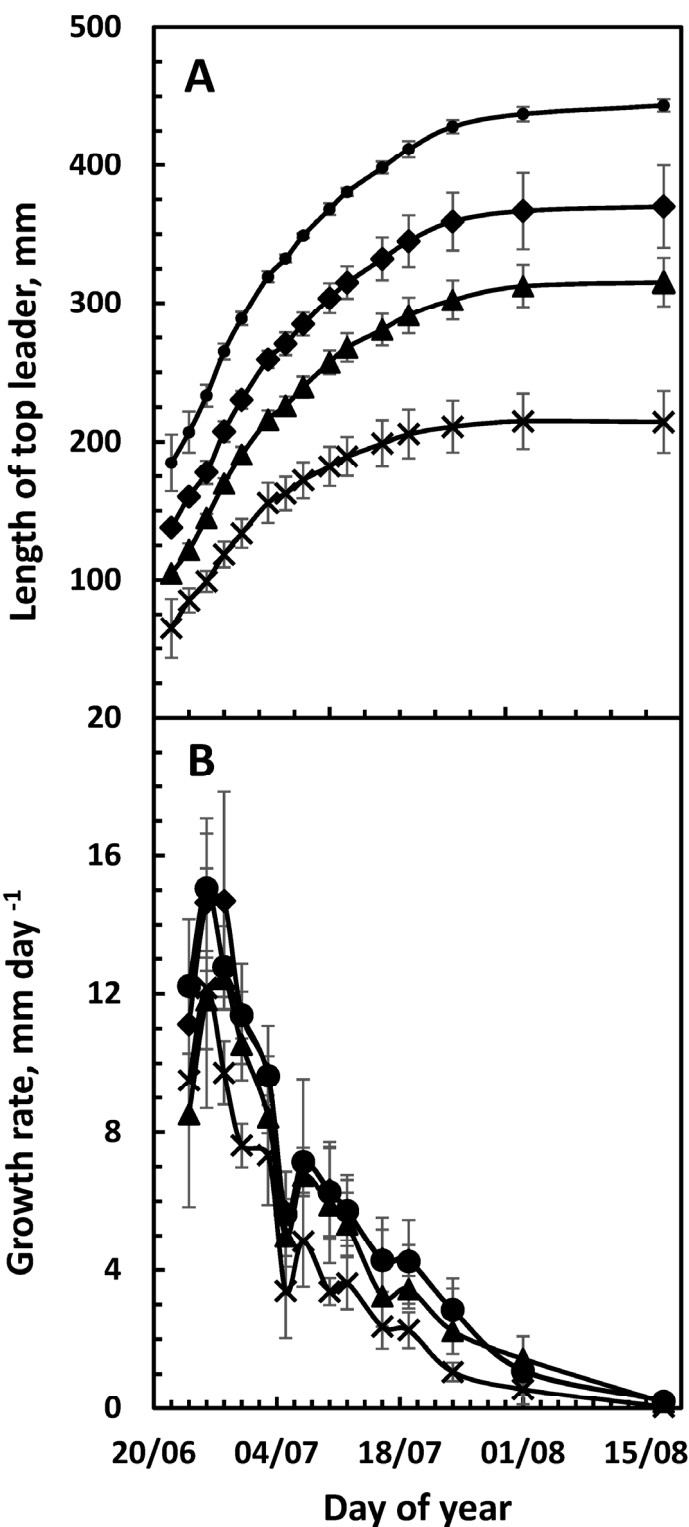

**Figure 4.** Growth of top-leaders of selected length. The top-leaders were selected on 22 June for being 5 (x), 10 (▲), 15 (♦) or 25 (●) cm tall. The length (**A**) and the daily growth rate (**B**) of the top-leader was followed until 18 August. Data are given as means of 3 trials ± SD.

*3.2. Daily Growth Rates*

Calculating the daily growth rate obtained in the years 2016 to 2022 showed that about 2 weeks after elongation had begun, the daily growth rate reached a rate of 12 to 22 mm

per day between 19 and 28 June, except for the year 2018, when the growth rate peaked on 2 June at 17 mm per day (Figure 1B, Table 2).

**Table 2.** Growth characteristics of *Abies nordmanniana* top-leader. Data are given for the years 2016 to 2022. Different letters denote significant differences between Students *t*-test.

| | | Maximum Growth Rate | Length of Top-Leader on Day of Maximal Growth Rate | |
|---|---|---|---|---|
| Year | Date | mm Day$^{-1}$ | mm | As % of Final Length |
| 2016 | 28 June | 12.0 [a] | 233 [a] | 45 |
| 2017 | 19 June | 16.8 [b] | 279 [a] | 52 |
| 2018 | 4 June | 14.6 [a] | 210 [a,b] | 39 |
| 2019 | 20 June | 13.7 [a] | 258 [a] | 58 |
| 2020 | 20 June | 22.5 [c] | 243 [a] | 58 |
| 2021 | 27 June | 14.5 [a] | 340 [c] | 55 |
| 2022 | 28 June | 13.2 [a] | 204 [b] | 55 |

During the next 6 weeks, growth decreased at a low rate (Figure 1B). Although the daily growth rate for the entire top-leader declined slowly after about 20 June when monitored for the entire top-leader, data from individual sections showed a different picture. The daily growth rate of the individual four sections of the top-leader revealed a plasmatic development in the growth of the top-leader. Growth ceases from the base upward, and in sections 1 and 2 the maximal growth rate did not exceed 2 mm per day, while a rate of about 8 mm per day was obtained in section 4. Furthermore, growth cessation seems to move upward, in the same way as a wave that is slowing down. In sections 1–4, growth stopped on 25 July, 29 July, 15 July and 6 August, respectively (Figure 2B).

When the top-leaders of the early- and late-bud-bursting plants were analyzed, the growth data showed that their growth rate differed only significantly in the time period up to the time the maximal growth rate was reached (Figure 3B), while in the period that followed there were no significant differences observed in the daily growth rate, indicating that growth in the upper part of the top-leader was independent of the time of bud burst.

### 3.3. Growth and Climatic Correlations

The data obtained and presented in Figures 1–3 indicate that the growth before and after the maximum daily growth rate is reached is controlled differently. When growth characteristics for the years 2016 to 2022 were correlated with climate data from the previous year, the only significant correlation was between the average temperature in July and the maximum daily growth rate in the following season (Figure 5, Table 1) indicating that it is the development in the previous year that determines the growth potential of the top-leader. None of the climate data for the previous year could be correlated to the end-of-season length of the top-leader, and neither could the maximum daily growth rate obtained in the years 2016–2022 ($p = 0.51$) (Figure 6). When climatic data were correlated with the growth of the top-leaders after the maximum growth rate was obtained, the data revealed a strong correlation between total precipitation in the period 20 June to 1 August (r (4) = 0.91, $p = 0.01$) (Figure 7), showing that precipitation in this period was significant for the ability of top-leaders to express the full growth potential. However, a detailed analysis of the daily temperature and precipitation in the 2020 season and the growth rate the following day did not show any significant correlation, although the daily growth rate and the maximum daily temperature revealed similar curves (Figure 8).

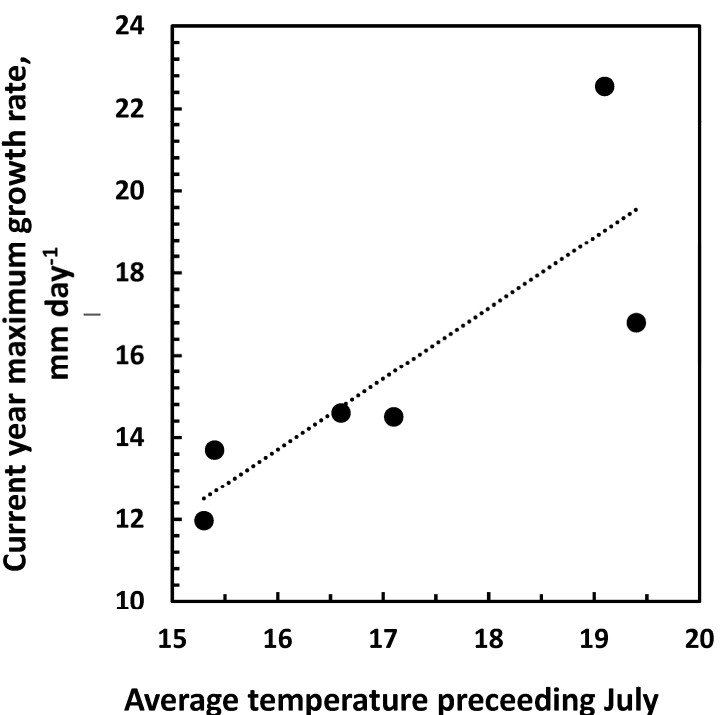

**Figure 5.** Correlation between the average temperature in July the previous year before top-leader elongation, and the maximal daily growth rate. Data are based on data from the years 2016 to 2022. Correlation: $R^2 = 0.67$; $p = 0.045$. $r(4) = 0.82$, $p = 0.045$.

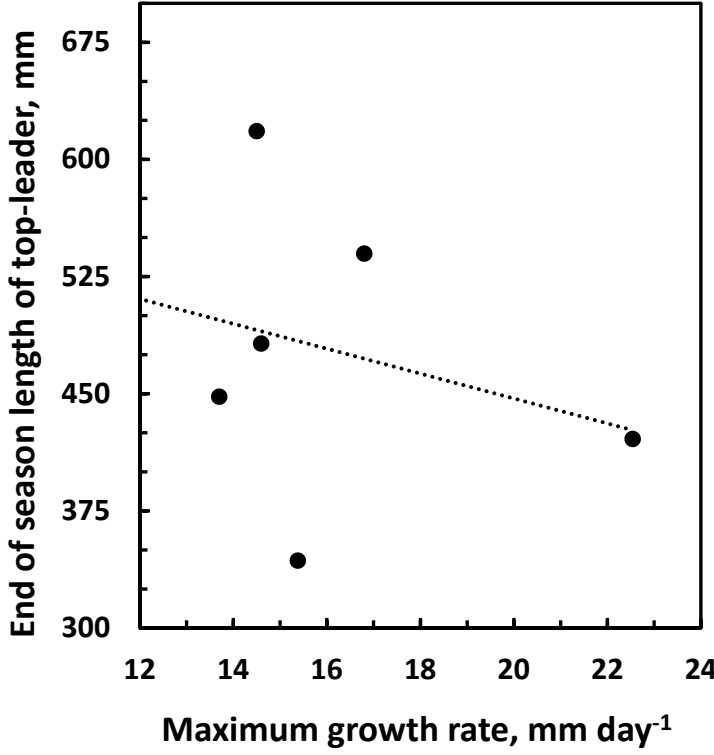

**Figure 6.** Correlation between the maximum daily growth rate and the length of the top-leader at the end of the season. Data are from the years 2016–2022. $r(5) = -0.30$, $p = 0.51$.

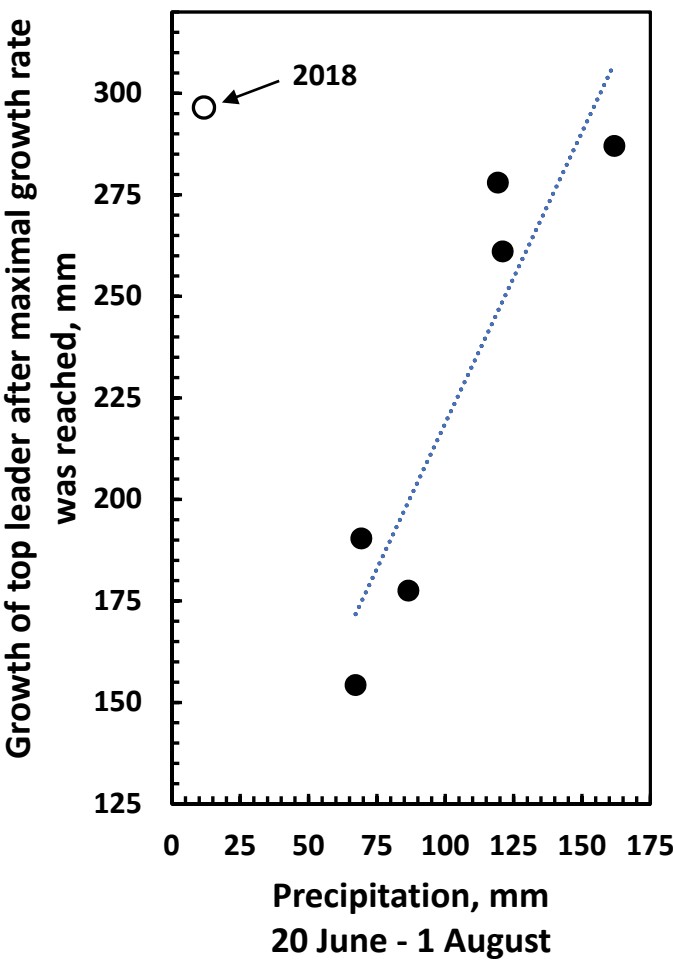

**Figure 7.** Correlation between the July precipitation and the growth of the top-leader after the maximal daily growth rate was reached at the end of June (see Table 2). Data are for the years 2016–2022. $r(4) = 0.91$, $p = 0.01$. Data for the 2018 season are omitted in the statistics because time of growing was significantly different to the other years (see Figure 1A). The 2018 data is shown as (○) in the figure.

In *Abies nordmanniana*, all stem units appear at bud break, unlike dicotyledon trees, where nodes appear and elongate successively [23]. In Denmark, the bud break in *Abies nordmanniana* normally occurs in late May, and stem elongation appears as a typical sigmoid growth curve, similar to that in observations of the balsam fir [13]. It is normally the entire length of the top-leader that is determined when growth is monitored [24]. Doing this monitoring over a 6-year period revealed a very stable growth pattern (Figure 1A). After a slow initial growth for the first two weeks, the daily growth rate increased to a maximum of 18 mm a day (Figure 1B), considerably greater than the 4.6 mm that was the highest growth rate observed by Kozlowski and Ward in five different conifers [13]. The maximum growth rate was obtained within a week around 24 June, where the top-leaders reached half the final length (Table 2), and occurred after 3 weeks out of a total of 9 weeks of growth (Figure 1B). In angiosperms, the maximal daily growth rate was normally observed in the middle of the growth period [25]. The growth thus appears to be asymmetric, and may be regulated differently before and after the maximum daily growth rate was observed. As previous studies had shown that growth did not cease uniformly in the top-leader [23], the top-leaders were divided into four sections, and their growth was monitored. It became evident that the observed growth pattern consisted of two phases. The first phase lasted from the bud break to about 24 June, when the maximal daily growth rate was registered.

The second growth phase began after the maximum growth rate was reached, and lasted until the end of the growing season, approximately 1 August.

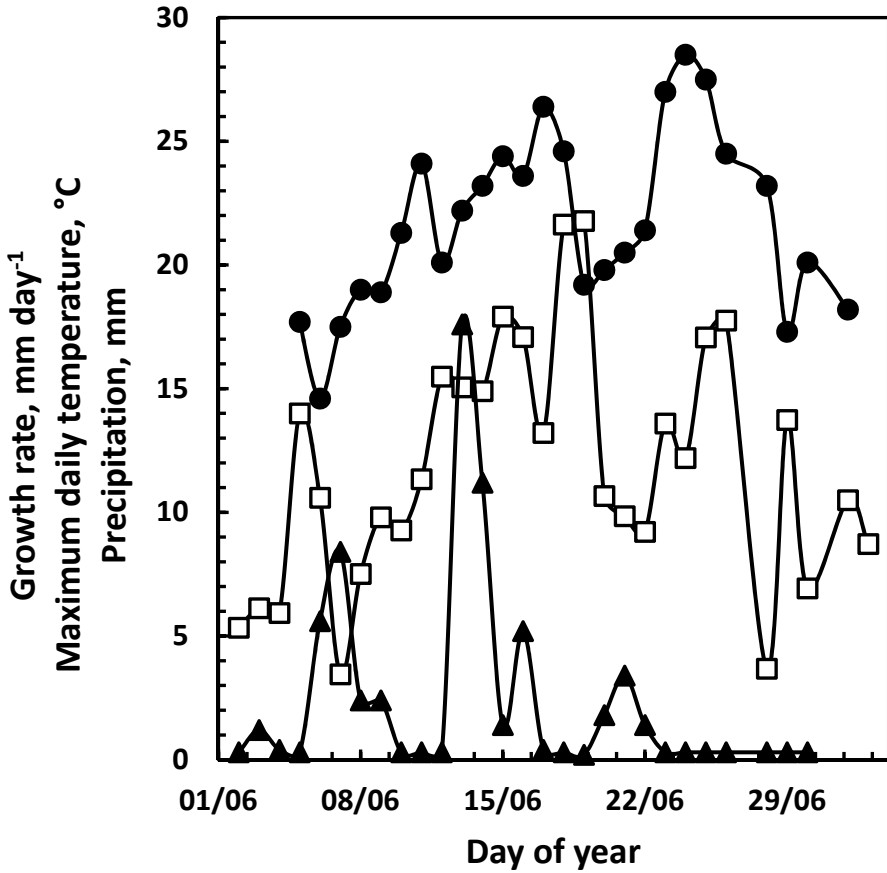

**Figure 8.** Top-leader growth and climate data in 2022. Daily growth rate of the top-leaders was determined daily (□), concurrently with the maximum daily temperature (●) and precipitation (▲) in the same period. No significant correlation was found between the daily growth rate and the climate data.

## 4. Discussion

At the beginning of the first growth phase, a substantial export of minerals from the old to the new developing needles occurs [26], which could be an additional source of the osmotic pressure driving the fast daily growth rate after bud break. This would also explain the long epidermal cells observed in section 1 [23]. Growth in section 1 ceases shortly after the bud break (Figure 2B), and top-leaders which have a late bud break are shorter than those who have an early bud break (Figure 3A), and they have the shortest epidermal cells [23]. The observed difference in the daily growth rate of buds that have an early or late bud break (24 May vs. 7 June) (Figure 3B) might be caused by less mobilization of minerals from old needles, thus generating less osmotic pressure to facilitate cell elongation. The ability of the trees to mobilize minerals from the old needles may therefore be a possible genetic factor that might be incorporated into breeding programs.

At the time when the second growth phase begins, cessation of growth begins at the basis of the top-leader (Figure 2A,B, [23]). The maximum daily growth rate increased in an acropetal wave and peaked in the upper section on 11 July at 8.5 mm per day (Figure 2B), which was 22 days after the maximum daily growth rate peaked in the entire top-leader (Figure 1B). After mid-June, the distance between the needles increases, and the level of cytokinins and auxin also begins to increase [27]; this could be a result of the continued cell division that must occur in the top section, as the length of the epidermal cells is shorter in section 4 than in section 1 [23], but section 4 contributes about 50% of the final

length. However, while the internode growth in arabidopsis is due to activity in the rib meristem [28], and the intercalary meristems in grasses [29], the origin of meristematic activity within the elongating stem of the Nordmann fir has not been determined.

The exception to the growth data was the year 2018, which was extremely warm in Europe, and which influenced tree growth in the following years [30]. In 2018, the average temperature in Denmark was 1.7 and 3.8 °C warmer in April and May, respectively, than the average during the years 2016 to 2022 (Table 1). This caused the top-leaders to start growth 2–3 weeks earlier than the average in the years 2016 to 2022, causing the date of maximal growth to be reached 19 days earlier than the average (Figure 1B). However, the maximum growth rate and the length of the top-leader at the end of the growth season in 2018 was similar to the years 2016–2022 (Figure 1A and Table 2). Although it was surprising that the elongation of the top-leader was not affected by warm and dry climate conditions, this was also observed in Norway spruce by Hayatgheibi et al. [30]. Their observations showed that an influence on growth occurred in the following years. In this work, no growth reduction was observed in the following years, probably due to the criteria for selecting trees for the measurements, as different trees with equal growth were selected each year. Therefore, the warmer temperatures in April and May 2018 only caused an earlier bud burst of the entire population, similar to that in observations of the Norway spruce [30], without influencing the daily growth rate. Thus between years, the time from bud burst to the maximum daily growth rate is reached seems to be constant (Figure 1A,B), contrary to early- and late-bud-breaking trees within a population, where a late bud break decreased the time to when the maximal growth rate was reached (Figure 3B).

To vertify that the time of bud break determines the final length of the top-leader, growth data from plants having early or late budbreak (24 May vs. 7 June) was extracted from the 2020 series. Where the top-leaders of the early-bud-bursting plants reached a final length of 562 mm, the late-bud-bursting plants only reached 299 mm (Figure 3A). When the daily growth rate was determined, it showed that the difference in the growth rate between the two selections was only significant until the time the maximum growth rate was reached (Figure 3B). The daily growth rate thereafter was identical, indicating that factors from the apical meristem may support growth in the second phase.

To verify that the daily growth rate after 20 June was independent of the length of the leader at this time, top-leaders 5, 10, 15 or 20 cm long were selected on 22 June in 2022. The data confirmed that although the final length of the top-leader depended on the length of the top-leader on 22 June, the daily growth rate was identical in all samples (Figure 4A,B), supporting the fact that hormones from the developing apical bud determine cell division in the top section of the top-leader [27].

In plants with a preformed shoot initial at bud breake, the temperature in the previous growth season is important for the size and growth of the top-leader in the following year [31,32]. When climate data were obtained in this experiment from June to November in the years 2016 to 2022, the only significant correlation was not to the final length of the top leader, but between the maximum growth rate of the top-leader and the temperature in July of the previous year (Figure 5). Similarly, Junttila [11] observed that the number of stem units in the pine was correlated with the temperature of the previous year, and the elongation was positively correlated with the number of stem units, confirming that the growth potential depends on the temperature when the shoot initials are formed. Although the maximum daily growth rate for the 6-year period ranged from 12.0 mm in 2016 to 22.5 mm in 2020, it was not significantly correlated to the final length of the top-leader ($r(5) = -0.30$, $p = 0.51$) at the end of the season (Figure 6). The only significant correlation obtained was between precipitation from 24 June (day of the maximum daily growth rate) to 1 August (when growth had ceased) and growth in the same period. The correlations between drought and tree growth are well documented [33]. As the final length of the top-leaders depended on precipitation and this varied between the years (Supplement Tables S1–S5 for climatic data), the ability of the top-leader to fulfil the growth potential varied for individual years.

Although the maximum temperature of the previous 24 h of the daily growth rate followed the same trend, the data were not significantly correlated. In addition, daily precipitation did not appear to influence the daily growth rate (Figure 7). Neither the general climate conditions in the growth season nor the daily fluctuations in temperature or precipitation seem to influence the growth of the top-leader.

## 5. Conclusions

The elongation of the top-leader in *Abies nordmanniana* consists of two different types of growth: an early growth phase that consists mainly of cell elongation, whereby a growth rate of up to 22 mm per day may be reached, and a second, slower growth phase, which involves cell division. The difference between small and long top-leaders is determined by the growth rate in the first weeks after bud break, after which all top-leaders grow at the same daily growth rate The growth potential of top-leaders of *Abies nordmanniana* is determined by the temperature in July in the season where the shoot initials are formed. The precipitation in late June and July in the growth season influences the final length of the top-leader. The daily growth is not influenced by the climate conditions in the preceding 24 h.

**Supplementary Materials:** The following supporting information can be downloaded at https://www.mdpi.com/article/10.3390/f14061214/s1, Tables S1–S5 on the climatic data (Table S1 maximum temperature, Table S2, minimum temperature, Table S3 precipitation, Table S4 sunshine hours, and Table S5 on drought index).

**Funding:** This project was supported by a grant from the Danish Agricultural Agency (Grant 34009-15-0964) to B. Veierskov.

**Acknowledgments:** The author want to thank Steen Sørensen and HP for providing the trees used in this investigation, and Steen Sørensen for sharing knowledge of Christmas tree production.

**Conflicts of Interest:** The author declares no conflict of interest.

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
