# Peer review of "Top-Leader Growth in Nordmann Fir (Abies nordmanniana)"

_forests, doi:10.3390/f14061214_

Round 1

Reviewer 1 Report

The manuscript is discussing a long term investigation the effect of the climate on the tree growth and development. The introduction need more recent literatures .The manuscript is well-written but some comments need to be addressed:

- line 38. delete 'late'

- line 51. delete ]

- line 91. leave space between the number and the unit 5cm to be 5 cm

- line 106.  add space here June22nd, and  line 124

- line 129. delete the comma in "and,270", delete "and"

- Figure 1. figures and figure 3 caption . correct as in the pdf

- figure 5 is on the top of figure 4. fix and correct the capitation

- line 324. remove the space

- line 325. "by" should be "by"

- Table 1. Growth (mm)/day fix. length (mm)

- line 336. remove (

- line 340 add space after number

- line 363. no space  ° C

- line 425. remove :

Author Response

I have made a revision to the introduction and discussion, and are now showing the growth to the top leader the individual years in Figure 1. I have incorporated your comments in the revised manuscript.

Reviewer 2 Report

Dear author,

this study focus on the characteristic of top leader elongation of Abies nordmanniana during 6 years. The effect of prevision and current year climate on shoot elongation was investigated as well as the growth of basal and top part of leader shoot was studied. Although the results could be partly use to produce quality Christmas trees, there are not many new findings in this paper.  The novelty of this study should be better described.   Moreover, methods and data analysis are not sufficiently describe. Overall, this study brings interesting results, but the present study need considerable revision.

See my detail comments bellow.

Title: It has to be change as it is same as Martens et al. 2019, where author of this paper was also involved.

Latin name - always in italic and whole Latin name has to be given when it is mentioned for the first time - i.e. with author

 line 8-13 - Not new findings - see your article Martnes et al. (2019)

line 11-12 - English style

line 12 and 15 - How these correlations go together? This should be explain in abstract.

line 19-20 - Not clear

Keywords - other keywords should be used - not mentioned in title. Did you study stem growth?

Overall, Introduction is rather short and other references focus on bud and shoot development could be use. For example, studies focus on the correlation between climate and shoot growth are missing. Even though, author wrote that information about shoot elongation are missing since 1964 (line 32), it is not true. There are several papers, even for Abies normandiana, which were focus on bud and shoot development. I will also enlarge Introduction to identify novel finding in this study.

line 29-30 I don´t agree that stem ontogenesis is more complex than in roots.

line 33 - see Martes et al. (2019)

line 44-45 - Not clear. Can you describe it better?

line 28-62

line 74-80 - Not written what type of climate data were collected and used for analysis.

line 83-91 - Not clear paragraph. It is not obvious how many trees were used for analysis each year. They were each year 8-year old? Tree division in lots is not clear.

line 99-106 - Also not clear part. You should rewrite it to be sure that anybody can clearly follow your experiment design and repeat your study.

line 109-113 - Not clear. It is not clear from this description how your data were analysed. Did you use average data for each year or use individual data? How was "group" identified?

line 124-130 - Section 1 -  It is situated once on the base and next it is on the top. You should be consistence and "section 1" has to be always on the same position.

line 128-130 - Not clear - Section 1, 2, 3 and 4 contribute ...

line 139-141 - I don´t see this peak on Fig. 1A

line 144 - Valid only for year 2018

line 148-149 - This is true only for the upper part - section 4

line 159-160 - Not clear

line 169-174 - Final length of the top leader or the length of top leader after the maximal growth rate was reached? Should be Fig. 7.

line 174-176 - How July precipitation could influence leader growth when already 95% of entire leader is formed in July.

line 333-334 - English

line 338-339 - Can you discuss with results  Martens et al. 2019? Nothing novel.

line 341-342 - Not shown on Figure.

line 363-364 - The selection of year 2018 from analysis is not clear. Even though April and May temperature were higher they were similar in June and July.

line 409-411 - Not clear

Figures - You have to use the same day of year description on x-axis

Fig. 2 and 3 - Description of panel A and B is missing.

Fig. 4 - It is hidden by Fig. 5

Figure 8 - Not clear description in legend

You should rebuild your Figures - One Figure, which will be focus on leader elongation and than one Figure focus on daily growth rates.

Author Response

The title has been changed and I have focused more on Christmas tree production in the introduction to emphasise the need to get basic knowledge about growth. Although there is plenty of literature on climate and general plant growth and productivity, as well as bud development, I do not find much relating to stem elongation.

I have omitted Figure 6B, as it is too easy to compare the data to figure 1 and then come to the same conclusion as you. However, Figure 4A shows that these plants 1/3 of the growth occurs in July. The reason for this is that they were selected according to a different criterion than all the others. This has now been specified in the Material and Methods. This has also caused a minor change to Figure 7 as instead of using July precipitation, I now are using precipitation from June 24 where the maximal growth rate was reached. This alteration has actually strengthened the correlation slightly.

I find that re-doing the figures as suggested would also require a re-composition of the entire manuscript, which I feel should have a specific purpose to strengthen some specific part.

I have incorporated your other comments in the revised manuscript.

Reviewer 3 Report

1. The scientific name of the species should always be written in italics, in several parts of the document this is not observed, correct it.

2. In the 2.5. Statistical Data Analysis (Methods), it is necessary to better explain how climatic variables are related to growth, nothing is mentioned about this.

3. Explain why only climatic data from April to November were used, why not work with data from January to December.

4. It would be important to show a graph of growth measurements for each year (2016 to 2022), this would allow comparing the variability in growth per year and how this variability is related to climatic variables (precipitation and temperature).

5. If there were graphs of the relationship between monthly average growth and climatic variables (precipitation and temperature), it would be easier to see which variable and which month are the most important for growth.

6. Likewise, a graph with the correlations of the influence of the climatic conditions of the previous year would be interesting.

7. There is a lack of further discussion on the influence of climatic variables (precipitation and temperature) on growth.

8. You indicated, to obtain plants of equal developmental size each year, measurements were made on trees grown on two different commercial Christmas tree farms. Is there a difference between the growths of the two farms?

9. You indicated, the trees were fertilized with a commercial fertilizers of 50 kg N ha (NPK 23-3-7 + Mg), split in two, with two thirds in April and one third in July. Does this represent a significant effect on growth compared to applying nothing? Nothing is said about this.

10. Figures 4 and 5 are superimposed and cannot be seen well. Correct them

Author Response

As the climatic factors have influence on all the biochemical factors that are related to growth, I find it difficult to select just some because none biochemical measurements have been performed.

I have now included all the month in Table 2. As you pointed out, the winter month might also be of interest to the readers

I have made the suggested alteration in Figure 1A

I think it would generate too many figures presenting all the correlations. However, since the data are presented, the reader may do so himself.

The two farms were chosen because the tree growth was similar in both places. As for fertilization, then all growers do fertilize according to the recommendation of the Danish Christmas Growers Association. I have now included a reference for their fertilization experiments.

Your other comments have been incorporated into the revised manuscript.

Round 2

Reviewer 2 Report

Dear author,

the manuscript was significantly improved, but it still need to improve the Figures description.

see my previous comments:

Figures - You have to use the same day of year description on x-axis

Fig. 2 and 3 - Description of panel A and B is missing.

Fig. 4 - It is hidden by Fig. 5

Figure 8 - Not clear description in legend

Author Response

The figures have been updated. I do not know why fig 4 is hidden by figure 5 in your version. I have now tried to delete them and put them in once again and hope that has helped.

Reviewer 3 Report

I have no more suggestions

Author Response

The figures have been updated